# Feasibility and Accuracy of Noninvasive Continuous Arterial Pressure Monitoring during Transcatheter Atrial Fibrillation Ablation

**DOI:** 10.3390/jcm12062388

**Published:** 2023-03-20

**Authors:** Andrea Di Cori, Matteo Parollo, Federico Fiorentini, Salvatore Della Volpe, Lorenzo Mazzocchetti, Valentina Barletta, Luca Segreti, Stefano Viani, Raffaele De Lucia, Luca Paperini, Antonio Canu, Gino Grifoni, Ezio Soldati, Maria Grazia Bongiorni, Giulio Zucchelli

**Affiliations:** Second Division of Cardiology, Cardiac-Thoracic and Vascular Department, University Hospital of Pisa, 56126 Pisa, Italy

**Keywords:** atrial fibrillation ablation, arterial pressure, blood pressure, ClearSight, noninvasive continuous arterial pressure monitoring

## Abstract

Introduction: Transcatheter atrial fibrillation (AF) ablation is still carried out with continuous invasive radial arterial blood pressure (IBP) monitoring in many centers. Continuous noninvasive blood pressure (CNBP) measurement using the volume-clamp method is a noninvasive alternative method used in ICU. No data on CNBP reliability are available in the electrophysiology lab during AF ablation, where rhythm variations are common. Background: The objective of the present study was to compare continuous noninvasive arterial pressure measured with the ClearSight device (Edwards Lifesciences, Irvine, CA, USA) with invasive radial artery pressure used as the reference method during AF ablation. Methods: We prospectively enrolled 55 consecutive patients (age 62 ± 11 years, 80% male) undergoing transcatheter AF ablation (62% paroxysmal, 38% persistent) at our center. Standard of care IBP monitoring via a radial cannula and a contralateral noninvasive finger volume-clamp CNBP measurement device were positioned simultaneously in all patients for the entire procedure. Bland-Altman analysis was used to analyze the agreement between the two techniques. Results: A total of 1219 paired measurements for systolic, diastolic, and mean arterial pressure were obtained in 55 subjects, with a mean (SD) of 22 (9) measurements per patient. The mean bias (SD) was −12.97 (13.89) mmHg for systolic pressure (level of agreement −14.24–40.20; correlation coefficient 0.84), −1.85 (8.52) mmHg for diastolic pressure (level of agreement −18.54–14.84; correlation coefficient 0.77) and 2.31 (8.75) mmHg for mean pressure (level of agreement −14.84–19.46; correlation coefficient 0.85). Conclusion: In patients undergoing AF ablation, CNBP monitoring with the ClearSight device showed acceptable agreement with IBP monitoring. Larger studies are needed to confirm the potential clinical implications of continuous noninvasive BP monitoring during AF ablation.

## 1. Introduction

Catheter ablation for atrial fibrillation (AF) has emerged as an important rhythm-control strategy and is by far the most common cardiac ablation procedure performed worldwide. Despite the fact that many technological advances have been incorporated into clinical practice in order to improve success rates and minimize complications, hemodynamic status monitoring remains mandatory to monitor potential ablation complications such as anesthesia-related hypotension and/or cardiac tamponade [1]. Noninvasive blood pressure measurement (NIBP) using an oscillometer arm cuff, even if practical, is unable to detect sudden arterial blood pressure changes [2]. Nevertheless, continuous invasive arterial (radial or femoral) blood pressure (CIBP) monitoring, still used in many centers, is inevitably associated with patient discomfort, procedure prolongation with unnecessary room occupation, and potential local vascular complications [3].

ClearSight™ (CS) (Edwards Lifesciences, Irvine, CA, USA) is a noninvasive hemodynamic monitoring system that uses a digital cuff and “volume clamp” technology to obtain a continuous arterial waveform. The system measures continuous noninvasive blood pressure (CNBP) with an improved method based on the Penaz and Wesseling studies [4,5] through an arm cuff integrated with a photoplethysmograph and applied to the patient. The cuff pressure is automatically kept between systolic and diastolic blood pressure values, then CNBP is measured indirectly from the middle phalanx of one finger using the volume-clamp method.

Although the CS has been validated and investigated in different settings, including critical patients, ref. [6] surgical patients for perioperative monitoring [7,8,9], and transcatheter aortic valve replacement [10], no data are available in patients undergoing transcatheter AF ablation, whereas not only blood pressure but also cardiac rhythm variations are often observed.

The aim of this study was to investigate the feasibility and accuracy of the CNBP in comparison with the current gold standard CIBP in patients undergoing transcatheter AF ablation.

## 2. Materials and Methods

We conducted a single-center prospective validation study from March 2021 to August 2021. Patients aged 18 years or more with an indication for transcatheter AF ablation (either paroxysmal or persistent AF) were enrolled. Radiofrequency ablation, as well as cryoablation procedures, were included in this study. The Declaration of Helsinki was adequately addressed, and the study was approved by the local ethics committee.

In all patients, the following clinical, echocardiographic, and procedural characteristics were collected: age, body mass index (BMI), left ventricular ejection fraction (LVEF), moderate-severe valvulopathy, hypertension (HTN), hyperlipidemia, coronary artery disease (CAD), diabetes mellitus (DM), and chronic kidney disease (CKD).

Standard of care CIBP monitoring was performed via sterile positioning of a 16 G radial cannula using the modified Seldinger technique under local anesthesia. For CNBP measurement, the ClearSight™ system was placed on the opposite arm from the IABP cannula, using the recommended finger cuff size, and placed on the intermediate phalanx of the first or second finger. (Figure 1A,B) After calibration of the heart reference system, measurement of systolic, diastolic, and mean arterial blood pressure was started. Values and continuous curves of IABP and CNBP were shown on the polygraph (Figure 1C,D, Appendix A), and systolic (SBP), diastolic (DBP), and mean (MBP) BP data were sampled and collected every 5 min. Data for which contemporary simultaneous measurements were unavailable (i.e., during trans-septal puncture or during the recommended 5-min-cuff release after 2 h of CNBP monitoring) were excluded from the analysis.

Descriptive statistics are reported as mean ± SD for normally distributed continuous variables or as a median with a range for skewed distributions. Categorical data are expressed as percentages, and differences in proportions were compared by χ^2^ analysis or Fisher exact test, as appropriate.

Agreement between measurements of BP measurements was assessed by Bland-Altman plot analysis [11,12]. To evaluate the accuracy and precision of the NCBP, a Bland- Altman analysis was performed for MBP, SBP, and DBP obtained in comparison with IBP. As a measurement of accuracy, the bias or mean of difference was used. The SD of the mean of the differences and 95% limits of agreement (mean of differences ± 1.96 SD) were used to assess precision. Correlation between CNBP measurements and IBP was also assessed by linear regression using the Pearson correlation coefficient, and concordance analysis was performed for MBP, SBP, and DBP. Statistical analysis was performed with the NCSS Statistical Software, version 2021 (NCSS, LLC. Kaysville, UT, USA).

## 3. Results

### 3.1. Study Population and Procedural Data

We enrolled a total of 55 consecutive patients with a prevalence of males (80%). The median age was 62 ± 11 years, with a heterogeneous cardiac risk profile and almost normal BMI. Regarding procedural indication, 62% of patients had a paroxysmal AF form, while 38% had a persistent form. Fifty-three procedures (96.4%) were conducted with the radiofrequency ablation technique and only two (3.6%) with cryoablation. According to the indication, the ablation strategy included PVI with additional substrate modification when required. The median overall procedural duration was 195.5 [IQ 163.5–318] minutes.

One major complication (cardiac tamponade) and three minor complications (vagal hypotension) occurred during the study period. No procedural-related deaths were observed.

Baseline patient characteristics and procedural results are shown in Table 1.

### 3.2. CNBP and IBP Measurements

A total of 1219 paired measurements for SBP, DBP, and MBP were obtained in 55 subjects, with a mean (SD) of 22 (9) and a median of 22 (25–75° interquartile range: 16–29) measurements per patient. Total measurements per patient ranged from 4 to 41. In the overall population, mean bias (SD) was −12.97 (13.89) mmHg for SBP (level of agreement −14.24–40.20; correlation coefficient 0.84), −1.85 (8.52) mmHg for DBP (level of agreement −18.54–14.84; correlation coefficient 0.77) and 2.31 (8.75) mmHg for MBP (level of agreement −14.84–19.46; correlation coefficient 0.85) (Figure 1). Percentage error for MBP, SBP, and DBP measured by CNBP amounts to 17.50%, 19.45%, and 21.76%, respectively.

A sub-analysis was also performed to assess agreement between CNBP and IBP during sinus rhythm and during atrial fibrillation. A total of 726 paired measurements for SBP, DBP, and MBP were obtained during sinus rhythm. The mean bias (SD) was 13.18 (12.92) mmHg for SBP (level of agreement −12.14–38.50; correlation coefficient 0.88), −1.93 (8.01) mmHg for DBP (level of agreement −17.63–13,76; correlation coefficient 0.80) and 2.32 (8.48) mmHg for MBP (level of agreement −14.29–18.94; correlation coefficient 0.88). A total of 455 paired measurements for SBP, DBP, and MBP were obtained during atrial fibrillation. The mean bias (SD) was 12.63 (15.38) mmHg for SBP (level of agreement −17.52–42.78; correlation coefficient 0.75), −1.72 (9.31) mmHg for DBP (level of agreement −19.98–16,54; correlation coefficient 0.72) and 2.29 (9.21) mmHg for MBP (level of agreement −15.76–20.34; correlation coefficient 0.77).

## 4. Discussion

The present study demonstrated for the first time the accuracy of the CNBP device during AF ablation, a special playground where many factors may account for multiple and unpredictable blood pressure changes.

Specifically, the main results of the study were:In patients with AF undergoing ablation, SBP, DBP, and MAP measurements monitored noninvasively with the CNBP were accurate and precise compared with invasive radial arterial catheter-derived measurements.CNBP results were accurate both in SR and in AF.CNBP results were accurate both under sedation and general anesthesia.

Although the safety, efficacy, and effectiveness of AF ablation are consistently improved in the last decade, [13,14] procedural management may be complex, and life-threatening complications continue to be observed [15]. Nevertheless, indications are expanding to ever sicker patients, such as those with advanced age, obesity, and cardiomyopathy [16,17,18].

Traditionally hemodynamic monitoring during AF ablation has been performed via invasive femoral or radial arterial access. However, IBP monitoring continues to account for procedural lengthening and potential access site complications [19,20]. As recently suggested, AF ablation, if performed under general anesthesia and supported by intra-cardiac echocardiographic (ICE) monitoring, seems to can be safely performed also with a noninvasive standard cuff blood pressure every 3 min [21].

Unfortunately, unlike in the USA, where it is usually performed under general anesthesia with ICE monitoring, in Europe, conscious sedation without ICE is the predominant approach, [22] demanding more comfortable and reliable tools for BP monitoring.

The CNBP monitoring system has been previously studied and validated in multiple small-scale studies. The first Nexfin generation and, later, the Clearsight^TM^ demonstrated to be accurate and reliable in various clinical settings [23,24], such as non-cardiac surgery [24,25], cardiac surgery [26], and TAVI [10].

The originality of the present study is the special scenario studied. Indeed, AF ablation represents a heterogenous playground where many different factors may account for BP variations and measurements.

First, the AF ablation patient population may include many different sub-populations characterized by different volume-loading conditions, such as obesity and elderliness.

Secondly, the anesthesiologic regimes may space from slight sedation to general anesthesia, with different related risks of hypotension. Third, the concomitant anticoagulation needs to control hypertension minimizing any potential intracranial bleeding. Fourth, fluid injection and heart rhythm instability (including ectopies and irregular AF-related cycle length variations) may definitely be responsible for sudden BP change. (Appendix A) Finally, even if rare, the risk of cardiac perforation with tamponade makes mandatory reliable and fast BP feedback.

In line with previously published data, [24] our study confirmed that in patients with AF undergoing ablation, continuous SBP, DBP, and MAP measurements monitored noninvasively with the CNBP were accurate and precise compared with invasive radial arterial catheter-derived measurements. Correlations were confirmed during the overall procedural duration (where fluid loading is different), independently from the sedation regimen (general anesthesia vs. sedation) and from the baseline rhythm (no differences were observed during sinus rhythm or during AF).

Specifically, according to Bland-Altman analysis (Figure 2 and Figure 3), results were judged for good MAP and DBP and moderate for SBP, with good trending capabilities of the system (Table 2). A real-time BP evaluation offered the opportunity of managing hypo- or hypertension intra-procedurally. Immediate hypotension detection suggests excluding a cardiac tamponade and correcting the vascular volume, the anesthetic administration, or using vasopressors or inotropes. In our series, we reported only one case of cardiac tamponade, promptly diagnosed (after evidence of NCIP hypotension) and percutaneous treated under CNBP surveillance.

Intraoperative blood pressure is usually measured intermittently using noninvasive oscillometric devices every 3–5 min. Continuous monitoring facilitates early diagnoses of hypotension, thus potentially promoting timely treatment, especially when ICE monitoring is not routinely used. If intraoperative hypotension may cause cerebral hypoperfusion, which can result in hypoxia and subsequently cerebral ischemia, similarly intraoperative hypertension may lead to cerebral hyperperfusion with hemorrhagic stroke when not timely recognized and treated.

Recently, a new ClearSight™ system algorithm to predict hypotension has been developed based on machine learning that uses arterial waveform to predict intraoperative hypotension (Hypotension Predictor Index, HPI) [27]. Further studies should be carried out to analyze HPI quality and feasibility during EP procedures.

The results of this study should be interpreted considering some limitations.

It was a single-center prospective observational study with a relatively small sample size of patients undergoing AF ablation. Thus, results might not be generalizable to other clinical settings or extrapolated for other scenarios. In certain populations, such as critical ICU patients, patients under hemodialysis, or Reynaud syndromes, CS has been shown to lack accuracy and precision compared with the invasive arterial catheter, and thus, it cannot be recommended.

Similarly, according to the low sample size, the number of complications was very low, and any conclusions remain beyond the scope of the present study. Although a reduction of vascular complications commonly related to vascular access of IBP, larger studies should be carried out to further analyze CNBP in complications during EP procedures.

Another limitation of our study was the moderate density of coupled measurements obtained. Coupled data sampling was arbitrarily performed every 5 min, although, in a recent prospective observational study [26], authors concluded that the dynamics of the ClearSight™ course in comparison with IBP could be well reproduced even with a lower recording frequency.

Regarding data and correlations, the predefined acceptable range for bias (20%) and mean error (30%) were chosen based on previously published data. Nevertheless, the SBP was notably less accurate than the DBP and MAP. A small number of outliers was found; however, no complications or procedural variations from usual practice happened in those cases. Further study will evaluate the more accurate index to be used during the BP monitoring for AF ablation.

## 5. Conclusions

In patients undergoing AF ablation, noninvasive finger volume-clamp continuous BP monitoring with the ClearSight™ device showed acceptable agreement with standard of care invasive BP monitoring. Larger studies are needed to confirm these results and assess the overall safety of a noninvasive-only approach.

## Figures and Tables

**Figure 1 jcm-12-02388-f001:**
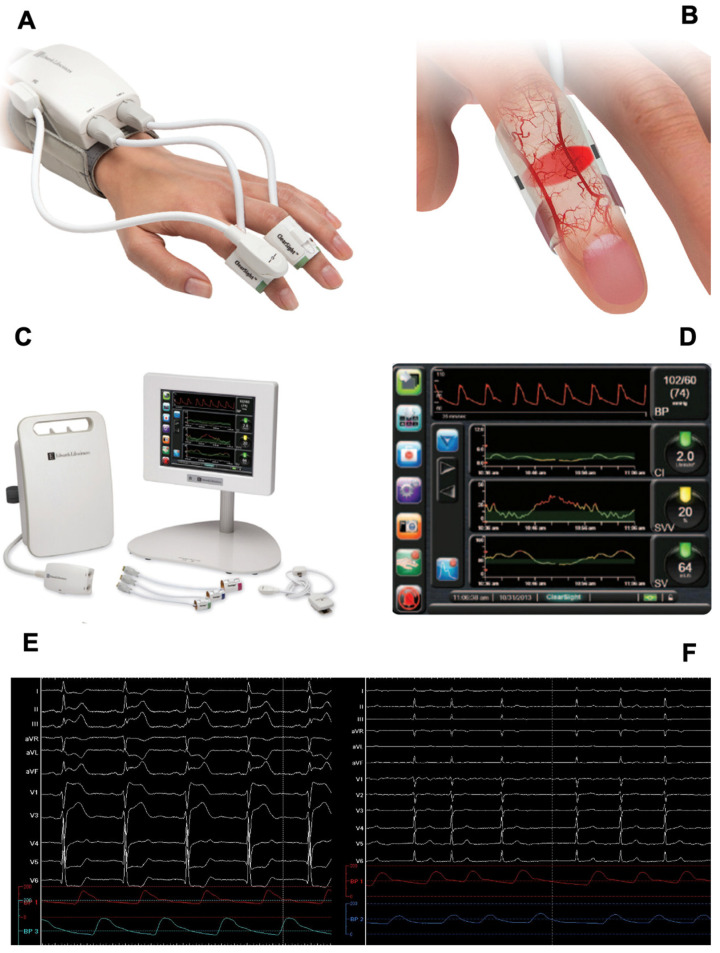
(**A**) ClearSight device when in use, with cuff placed on the first finger. (**B**) Representation of ClearSight finger-cuff device sampling digital arteries pulsation. (**C**) ClearSight monitor and patient hardware. (**D**) Screenshot of main ClearSight parameter view, showing CNBP tracing, cardiac index, stroke volume, and stroke volume variability live values and relative plots. (**E**,**F**) Live polygraph showing IBP (red line) and CNBP (blue line) curves alongside surface ECG, both during sinus rhythm (**E**) and atrial fibrillation (**F**).

**Figure 2 jcm-12-02388-f002:**
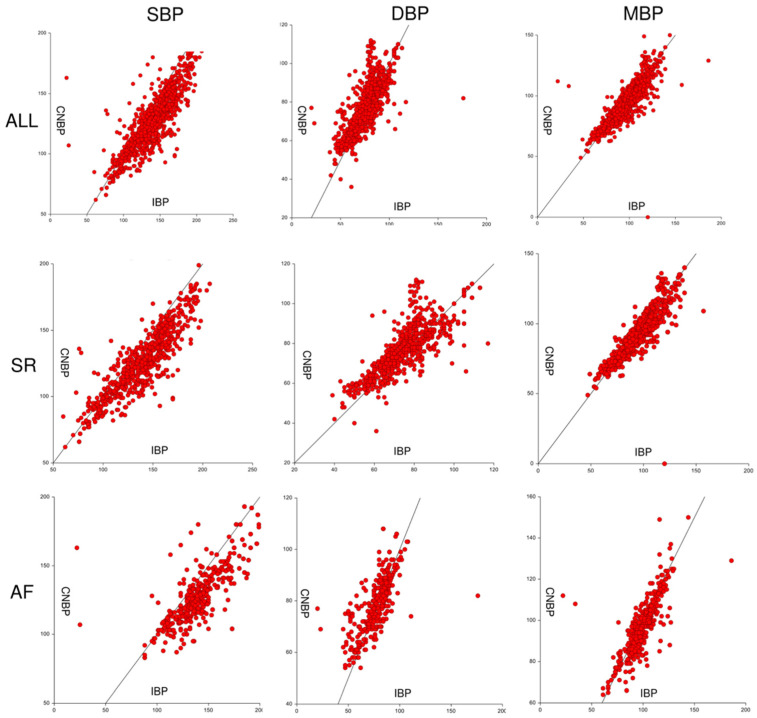
Correlation plots for systolic, diastolic, and mean blood pressure during sinus rhythm and atrial fibrillation. Values in mmHg. SR: sinus rhythm, AF: atrial fibrillation, SBP: systolic blood pressure, DBP: diastolic blood pressure, MBP: mean blood pressure, CBNP: continuous noninvasive blood pressure, IBP: invasive blood pressure.

**Figure 3 jcm-12-02388-f003:**
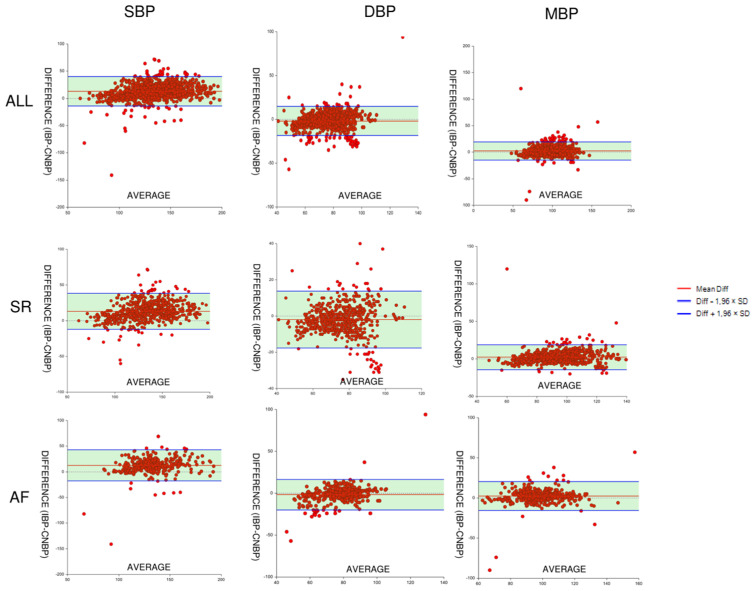
Bland-Altman agreement plots for systolic, diastolic, and mean blood pressure during sinus rhythm and atrial fibrillation. Values in mmHg. SR: sinus rhythm, AF: atrial fibrillation, SBP: systolic blood pressure, DBP: diastolic blood pressure, MBP: mean blood pressure, CBNP: continuous noninvasive blood pressure, IBP: invasive blood pressure, SD: standard deviation.

**Table 1 jcm-12-02388-t001:** Baseline patient characteristics and procedural results.

	N = 55
Male Sex, *n* (%)	44 (80%)
Age, years (mean ± SD)	62 ± 11
BMI (mean ± SD)	26.3 ± 3.7
LVEF, % (mean ± SD)	56.6 ± 8.2
Hypertension, *n* (%)	29 (52.7%)
CAD, *n* (%)	2 (3.6%)
Hyperlipidemia, *n* (%)	27 (49.1%)
DM, *n* (%)	3 (5.5%)
CKD, *n* (%)	2 (3.6%)
Moderate-severe valvulopathy, *n* (%)	7 (12.7%)
Paroxysmal AF, *n* (%)	34 (62%)
Persistent AF, *n* (%)	21 (38%)
Procedural time (minutes), (IQ)	195.5 [IQR 163.5–318]

AF, atrial fibrillation; CAD, coronary artery disease; CKD, chronic kidney disease; DM, diabetes; IQR, inter-quartile range.

**Table 2 jcm-12-02388-t002:** Mean and Standard Deviations of Arterial pressures from invasive and noninvasive measurements.

	Bias	SD	Correlation Co.
Mean Arterial Pressure	2.31	8.75	0.85
Systolic Blood Pressure	−12.97	13.89	0.84
Diastolic Blood Pressure	−1.85	8.52	0.77

SD, standard deviation.

## Data Availability

The data presented in this study are available on request from the corresponding author. The data are not publicly available due to privacy reasons.

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
