# Peer review of "Feasibility and Accuracy of Noninvasive Continuous Arterial Pressure Monitoring during Transcatheter Atrial Fibrillation Ablation"

_jcm, 2023, doi:10.3390/jcm12062388_

Round 1

Reviewer 1 Report

The authors compared non-invasive with invasive blood pressure monitoring during atrial fibrillation ablation, and found that the non-invasive measurement is sufficiently accurate.

1. The scenario of blood pressure measurement is rather specific. Blood pressure measurement is performed within minutes after a calibration. This is probably a very rare case for the use of the non-invasive continuous blood pressure monitoring. The results cannot be extrapolated to other senarios. 

2. The authors reported the number of study participants and the total and mean number of pairs. It would be useful to report the range of the number of pairs. If the number of pairs per participant is quite different, the authors may need to look at the accuracy by taking the mean of blood pressure per participant.

3. In the figures, the authors may need to report the number of participants and the number of pairs.

4.   There are a couple of extreme outliers. The authors may need to explore what happened to these outliers.

Reviewer 2 Report

In this work, Zucchelli and coworkers compared the standard method of invasive radial arterial blood pressure monitoring (IBP) with a new, non-invasive method (CNBP) for continuous monitoring during AF ablation procedures.

The new method consists in a volume-clamp technique, which was compared with IBP through the Bland-Altman analysis. The measurements were performed simultaneously on 55 patients undergoing cryo- or radio-ablation after paroxysmal or persistent atrial fibrillation.

The manuscript is limited by the small sample of patients tested, as correctly stated by the Authors. Nevertheless, the performances of the new, non-invasive system for blood pressure monitoring are assessed rigorously and in detail. Thus, after a careful check of the English language throughout the manuscript for typos (i.e. Table 2), I recommend publication of this work.

Author Response

Dear reviewer,

We thank you for the time spent reviewing our manuscript and for the useful feedback you provided.

As suggested, we reviewed the manuscript for typos (and we again than the reviewer for pointing them out) and resubmitted a corrected version.